# The Sound of Silence: How Silenced Chromatin Orchestrates the Repair of Double-Strand Breaks

**DOI:** 10.3390/genes12091415

**Published:** 2021-09-15

**Authors:** Apfrida Kendek, Marieke R. Wensveen, Aniek Janssen

**Affiliations:** Center for Molecular Medicine, Section Molecular Cancer Research, University Medical Center Utrecht, Universiteitsweg 100, 3584 CG Utrecht, The Netherlands; a.kendek@umcutrecht.nl (A.K.); m.r.wensveen@umcutrecht.nl (M.R.W.)

**Keywords:** constitutive heterochromatin, facultative heterochromatin, DNA-damage repair, double-strand breaks

## Abstract

The eukaryotic nucleus is continuously being exposed to endogenous and exogenous sources that cause DNA breaks, whose faithful repair requires the activity of dedicated nuclear machineries. DNA is packaged into a variety of chromatin domains, each characterized by specific molecular properties that regulate gene expression and help maintain nuclear structure. These different chromatin environments each demand a tailored response to DNA damage. Silenced chromatin domains in particular present a major challenge to the cell’s DNA repair machinery due to their specific biophysical properties and distinct, often repetitive, DNA content. To this end, we here discuss the interplay between silenced chromatin domains and DNA damage repair, specifically double-strand breaks, and how these processes help maintain genome stability.

## 1. Introduction

An essential condition for organismal life is the ability to maintain an intact genome. The eukaryotic genome is under constant pressure from damaging insults that break or chemically modify the DNA. One particularly dangerous type of DNA damage is a DNA double-strand break (DSB), which causes the DNA double helix to be completely severed. DSBs can be caused by endogenous processes, such as replication fork stalling or the production of damaging metabolites, or exogenous sources, such as X-rays or chemotherapeutics (reviewed in [1]). When repaired improperly, DSBs can result in defects ranging from small insertions and deletions at the repaired site to the formation of major chromosomal rearrangements, such as translocations, dicentric chromosomes, or chromothripsis. These structural changes can contribute to a variety of developmental diseases as well as tumorigenesis through the loss or gain of coding sequences, or the formation of aberrant fusion genes (reviewed in [2,3,4]).

Eukaryotes employ two main DSB-repair pathways: homologous recombination (HR) and classical non-homologous end joining (c-NHEJ) (reviewed in [5]). During HR, usually considered the safest choice of DSB repair, end resection at the DSB results in a single-stranded 3′ DNA end that invades and perfectly copies a homologous sequence to repair the break site. HR is mainly promoted during and following DNA replication when an identical sister chromatid is present, which can be used as a homologous template. c-NHEJ, on the other hand, is active throughout the cell cycle and is generally more error-prone. During c-NHEJ, the two ends of the DSB undergo limited processing and are directly religated, often resulting in small insertions and deletions at the repaired site. Although HR and c-NHEJ are the two main DSB-repair pathways, several alternative DSB-repair pathways, such as alternative end-joining or single-strand annealing, can be employed as well. The choice of DSB-repair pathway depends on several aspects, such as the cell type or cell cycle phase in which the DSB occurs, but also the nature of the DSB and its surrounding DNA sequences (reviewed in [5]).

The repair of DSBs occurs in the context of the complex surrounding chromatin environment and demands proper chromatin remodeling to detect, access, and process the DNA breaks. Indeed, work from a variety of model systems has revealed that chromatin remodelers and chromatin-modifying enzymes are necessary to properly guide DSB repair by locally altering the chromatin surrounding the break site (reviewed in [6]). One of the first chromatin events at DSBs is the phosphorylation of the histone variant H2A.X [7] (γH2A.X in mammals, γH2A in yeast, and γH2A.v in Drosophila), which can spread up to two Megabases around the break site [8] and initiates a multitude of downstream repair events (reviewed in [9]). In addition to γH2A.X, many other chromatin changes occur at the break site, such as the ubiquitination or acetylation of histones, which often serve as binding platforms for repair proteins (reviewed in [10,11]). Besides DSB-induced histone modifications, cell cycle-driven chromatin changes also influence repair by directly controlling repair pathway usage. For example, unmethylated histone H4 Lysine 20 (H4K20me0), which is abundant on newly established nucleosomes following DNA replication, is recognized by BRCA1-BARD1, an HR protein complex [12,13]. As such, H4K20me0 directly couples the replicative state of chromatin with repair pathway choice by stimulating HR repair with the sister chromatid. Together, these pre-existing as well as DSB-induced chromatin changes serve to promote the faithful execution of repair within the complex chromatin environment.

The eukaryotic nucleus is packaged into a plethora of distinct chromatin domains that have each acquired specific molecular and biophysical properties. These chromatin domains range from actively transcribed euchromatic regions to more compact, transcriptionally inert, heterochromatin domains [14]. Research from the past decade has revealed that this variety in chromatin properties demands a DSB response specifically tailored to each chromatin domain. For example, DSBs in actively transcribed chromatin regions specifically promote HR repair pathway usage [15] and depend on the removal of proteins required for transcription, such as RNA pol II [16,17] as well as loss of histone marks associated with active transcription [18]. On the other hand, regions of constitutive heterochromatin require striking movements of the DSB to the periphery of the heterochromatin domain or the nuclear periphery to repair the damage [19,20,21,22,23]. These studies highlight the importance of acquiring unique DSB repair responses in different chromatin regions. Silenced chromatin regions, such as pericentromeric heterochromatin, present a particular challenge to the cell’s repair machinery. Not only do these domains possess compact, phase-separated structures [24,25,26,27,28] that demand local chromatin changes to access and repair the breaks, their DNA also often consists of thousands of repetitive sequences (reviewed in [29]), which necessitates a coordinated repair response to prevent aberrant recombination between these repeats.

Here, we review our current knowledge on the DSB-repair response in silenced chromatin domains. We will predominantly focus on repair mechanisms in pericentromeric heterochromatin and will highlight several studies performed in other silenced regions, such as lamina-associated domains or facultative heterochromatin.

## 2. Constitutive Heterochromatin

The eukaryotic nuclear environment can be roughly divided into two types of chromatin: euchromatin and heterochromatin (Figure 1). Euchromatin contains many active genes and is associated with open chromatin structures. In contrast, heterochromatin, which was identified in liverwort a century ago, is relatively transcriptionally silent, encodes fewer genes, and has a compact, dense conformation throughout the interphase of the cell cycle [30,31]. The most prominent type of heterochromatin is constitutive heterochromatin (c-Het). c-Het is riddled with repetitive sequences [32] and is characterized by Histone H3 Lysine 9 di- and tri-methylation (H3K9me2/me3) (Figure 1c) [33]. In Drosophila, the three methyltransferases G9a, Su(var)3-9, and dSETDB1 (eggless) are the major histone methyltransferases that catalyze H3K9 methylation [34]. G9a mono- and dimethylates H3K9, while dSETDB1 and Su(var)3-9 act redundantly in H3K9 di- and trimethylation [35]. H3K9me2/3 provides a binding site for the chromodomain of heterochromatin protein 1a (HP1a, Su(var)205) [36,37,38]. HP1a homodimerizes through its chromoshadow domain [39], which, in turn, creates a binding platform for a variety of heterochromatin proteins, including Su(var)3-9 [40], thereby creating a positive feedback loop to facilitate heterochromatin spreading. Several HP1-like proteins have been described in Drosophila (HP1a–e) (reviewed in [41]). Although HP1a is the most abundant c-Het component, HP1b is also recruited to c-Het domains in addition to its euchromatic localization [42]. Su(var)3-9 and HP1 proteins were initially discovered in Drosophila genetic screens aimed at identifying suppressors of variegation (i.e., Su(var)) (reviewed in [43]), and were later found to be evolutionarily conserved from fission yeast to human. In mammals, three HP1 homologs have been described (HP1 α, β, γ) that can all be recruited to c-Het (reviewed in [44]). These HP1 proteins on their turn can recruit additional silencing proteins, such as the transcriptional repressor Kap1, to further establish and maintain c-Het structure.

c-Het is estimated to cover around 25–90% of eukaryotic genomes [45,46] and is enriched at pericentromeric regions as well as sub-telomeres [33] (Figure 1). Across species, the underlying sequence of pericentromeric DNA is dominated by satellite repeats and transposons. Satellites are short, simple repeats, while transposons are coding DNA sequences related to viruses that, when intact, can ‘jump’ and propagate in the genome. However, most transposon sequences found in eukaryotic genomes are fragmented and, therefore, inactive (reviewed in [29]). In interphase, c-Het sequences can localize to different structures within the three-dimensional nuclear space, such as the nucleolus or nuclear lamina, which serve as interaction hubs for silenced sequences [47] (Figure 1). This three-dimensional localization of heterochromatin is often dynamic and depends on the organism as well as cell type assessed. For example, c-Het in the diploid nuclei of Drosophila larvae usually forms one cytologically distinct nuclear domain that remains compact throughout interphase [36] (Figure 1B), whereas c-Het in other contexts, such as mouse embryonic fibroblasts, coalesces into multiple domains [48,49].

c-Het is essential for genetic stability by maintaining centromere- and telomere- structure, as well as in preventing deleterious expression of repetitive sequences [50]. In line with this, loss of canonical c-Het components results in tumor formation in mice [51] and is associated with ageing [52,53]. Interestingly, H3K9me2/3-enriched regions are associated with high mutation rates [54,55], and increased copy number aberrations [56] in cancer, indicating that these silent regions are particularly vulnerable to faulty DNA damage repair.

### 2.1. General Principles of DSB Repair in c-Het

Due to the repetitive nature of its DNA sequences, DSB repair within the c-Het domain can be a precarious event. Aberrant recombination with identical repeats present on non-homologous chromosomes can result in chromosomal aberrations often associated with cancer and developmental diseases [50,57,58,59].

Intriguingly, regardless of the model system used, research in the last decade has revealed that heterochromatic DSB repair involves distinct spatiotemporal mechanisms [19,20,21,22,23] (Figure 2 and Figure 3). Pioneering experiments in yeast revealed the movement of DSBs in the repetitive ribosomal DNA to outside of the nucleolus [60]. Strikingly, experiments using Drosophila cells [19,20] or tissues [22], as well as mouse cells [21,23,61], unanimously identified similar movements of c-Het DSBs to the heterochromatin or nuclear periphery. This movement is independent of the number or source of DSBs and appears to be a distinct feature of heterochromatic DSB repair, conserved from yeast to mammals.

The movement of heterochromatic DSBs has been hypothesized to have evolved to help move the damaged repeat away from its homologous repetitive sequences and thereby prevent aberrant recombination events [19]. This suggests that mobile heterochromatic DSBs depend on the HR pathway for their repair. Indeed, heterochromatic DSBs in mouse cells as well as Drosophila cells or animals were found to utilize HR [19,21,22,62]. In line with this, the inhibition of DNA-end resection, an initiating event for HR, prevents heterochromatic DSB movement in both Drosophila and mouse cells, indicating that HR initiation drives the movement of DSBs [19,21]. The depletion of the c-NHEJ protein Ku70 or Ku80 did not affect heterochromatic DSB kinetics in Drosophila cells [19], suggesting that c-NHEJ is not associated with DSB movement and may be used less frequently at c-Het DSBs in Drosophila cells. However, the sequencing of heterochromatic DSB repair products in Drosophila larval tissues did identify the use of both HR and c-NHEJ at I-SceI-induced DSBs, with c-NHEJ being the most prominent repair pathway [22]. This dominant use of c-NHEJ in larvae likely reflects the distribution of cell cycle stage, since the majority of cells in larval tissues reside in G1 [22] and therefore mainly employ c-NHEJ. In line with this, CRISPR-Cas9-induced DSBs in mouse chromocenters were also found to recruit c-NHEJ proteins in G1 cells, while mainly recruiting HR proteins in the S and G2 phases of the cell cycle [21]. DSB repair analysis in both Drosophila and mouse has also revealed the use of single-strand annealing (SSA) at c-Het DSBs, albeit at a relatively low frequency [21,22]. Together, these studies indicate that both c-NHEJ and HR can be used at DSBs in c-Het, with c-NHEJ repair predominantly being used in G1, while HR repair, as well as its associated DSB movement, is mainly limited to S and G2.

Strikingly, heterochromatic DSBs reveal clear spatiotemporal regulation of the early and late steps of the HR-repair pathway in Drosophila. Early steps, such as DNA-end resection and loading of ATRIP (ATR-interacting protein), which is recruited to resected DNA, occur within the c-Het domain. However, later steps, such as the binding of Rad51, a protein required for strand invasion and completion of HR, to the resected single-stranded DNA ends, only occur upon arrival of the DSB at the heterochromatin- or nuclear- periphery [19,21]. This spatial separation of early and late HR steps is thought to prevent Rad51-dependent homology search within the c-Het domain, and thereby preclude aberrant HR with identical repeats on non-homologous chromosomes nearby.

### 2.2. The Role of Canonical Heterochromatin Proteins in c-Het Repair

Studies in multiple model organisms have revealed that canonical heterochromatin proteins are essential to prevent genome instability [51,63,64,65,66,67]. For example, loss of the canonical heterochromatin proteins Su(var)3-9 or HP1a in Drosophila results in the formation of aberrant repair products at heterochromatic repeats [19,65]. In line with this, loss of HP1a and Su(var)3-9 in X-ray irradiated Drosophila cells results in defects in the spatiotemporal regulation of DSB repair in c-Het [19]. The presence of canonical heterochromatin factors was not only found to be essential to promote movement of DSBs to outside the c-Het domain, but also to prevent the accumulation of Rad51 at DSBs within c-Het. This indicates that heterochromatin components actively contribute to Rad51 exclusion from c-Het and thereby regulate ‘safe’ HR repair of repetitive DNA sequences outside the c-Het domain to prevent genome instability.

In contrast to Drosophila cells, loss of the canonical heterochromatin protein HP1α/β/γ or Kap1 in mouse cells does not lead to the accumulation of Rad51 inside the c-Het domain at CRISPR-Cas9-induced DSBs [21]. This difference between mouse and Drosophila c-Het, in their dependency on c-Het factors for Rad51 exclusion, could reflect species-specific differences. However, to conclude this, future experiments would have to rule out differences caused by the various DSB-inducing agents used in the different studies.

How the c-Het domain mechanistically excludes Rad51 from binding to DSBs remains unknown. However, it is tempting to speculate that the phase-separation properties of HP1a [25,68], one of the main c-Het constituents, are incompatible with Rad51 entering the c-Het domain. Future biophysical studies will have to elucidate if and how c-Het phase properties allow the accumulation of certain repair proteins in c-Het, such as the early repair proteins ATRIP or Mu2, while excluding late HR proteins, such as Rad51.

### 2.3. Local Chromatin Changes at c-Het DSBs

The specific dynamics of c-Het DSBs suggests that unique local chromatin changes are necessary for repair. Indeed, evidence from mammals indicates that several canonical heterochromatin proteins are directly phosphorylated by DNA damage kinases upon DSB induction in c-Het [69,70,71,72,73]. These phosphorylations are thought to stimulate heterochromatin relaxation at the break site and thereby promote repair (Figure 2). In mouse cells, the DNA damage kinase ATM directly phosphorylates the heterochromatin protein Kap1 [70,73]. This phosphorylation promotes the release of the nucleosome remodeler CHD3 and thereby allows c-Het relaxation [71]. Additionally, casein kinase 2 (CK2) has been found to phosphorylate HP1β, thereby promoting its release from heterochromatin and subsequent chromatin expansion at break sites [72] (Figure 2). Similar mechanisms have been described in *Arabidopsis thaliana*, where the heterochromatin-associated histone variant H2A.W.7 is phosphorylated by ATM to enhance DSB repair in chromocenters [74]. Indeed, in Drosophila, the DNA damage kinase ATR is also essential for the expansion of the c-Het domain as well as movement of DSBs to the heterochromatin periphery [19]. Together, these studies indicate that DNA damage kinase activities influence the local heterochromatin landscape to promote DSB dynamics in multiple organisms. Future studies aimed at creating a comprehensive overview of all c-Het proteins being phosphorylated upon DNA damage could yield important insights into how different kinases control c-Het conformation at DSB sites in space and time, and how this promotes repair.

In addition to the phosphorylation of canonical heterochromatin proteins, recent evidence also highlights a direct role for histone modifiers in heterochromatin repair. *Drosophila* KDM4A (dKDM4A) belongs to the jumonji family of lysine demethylases and can demethylate histone tails, such as the transcription-associated H3K36me2/3 as well as the heterochromatin-associated histone modifications H3K9me2/3 and H3K56me3 [75,76,77,78,79,80]. dKDM4A is recruited to c-Het through its direct interaction with HP1a and is highly enriched in the c-Het domain throughout the cell cycle [80]. Remarkably, dKDM4A plays dual roles in the c-Het domain: while it plays a non-enzymatic role in gene silencing and the maintenance of c-Het organization, its enzymatic activity is specifically required for the movement and repair of heterochromatic DSBs [80,81] (Figure 2). Upon DSB induction in c-Het, dKDM4A promotes the removal of the canonical heterochromatin marks H3K9me2/3 and H3K56me3 at the break site, which promotes timely repair by c-NHEJ [81]. Indeed, in the absence of dKDM4A, HR repair is enhanced at heterochromatic DSBs, which is alleviated by inhibiting the H3K9me2/3 methyltransferase Su(var)3-9 [81]. The dependency of heterochromatin repair on dKDM4A suggests that DSBs arising in c-Het require break-proximal chromatin changes to promote repair. Removing the heterochromatic histone marks H3K9me2/3 and H3K56me3 at c-Het DSB sites might promote chromatin relaxation that creates a permissive environment for repair protein binding and/or DSB movement outside of the phase-separated HP1 domain [25,81].

Interestingly, members of the human KDM4 family are often overexpressed in cancer [82] and both human KDM4A [83] and KDM4D [84] directly promote repair of DSBs. However, whether human KDM4 proteins specifically play a role in c-Het repair in human cells remains to be tested.

### 2.4. SUMOylation and the Nuclear Periphery in c-Het Repair

An emerging player in the DSB response is the modification of proteins by small ubiquitin-like modifier (SUMO) [85]. Recent evidence indicates that SUMOylation, through the action of the SMC5/6 chromatin complex, is also specifically required for c-Het DSB repair. SMC5/6 is a ring-shaped protein complex related to cohesin and condensin and is essential for the maintenance of genome integrity (reviewed in [86]). Loss of SMC5/6 in Drosophila results in defects in DSB movement to the heterochromatin periphery, as well as the accumulation of Rad51 at DSBs within the c-Het domain [19]. Although SMC5/6 plays numerous roles in the nucleus [86], one current prevailing hypothesis is that its main function in heterochromatin repair is to promote the addition of SUMO modifications to target proteins at the DSB site through its interaction with the SUMO E3 ligases Quijote and Cervantes (homologs of mammalian Nse2) [20,87]. SUMOylation of these target proteins is thought to prevent Rad51 accumulation within the c-Het domain as well as to promote movement of the DSB outside the heterochromatin domain [20,87] (Figure 3). This shares similarities with the repair of DSBs in the repetitive ribosomal DNA (rDNA) of budding yeast, which also depends on the activity of the SMC5/6 complex for DSB movement to the nucleolar periphery [60]. The conservation of the role of the SMC5/6–SUMO pathway in heterochromatin repair in mammalian cells remains to be tested. However, loss of SMC5/6 did result in a minor defect in DSB re-localization to the heterochromatin periphery and an increase in c-NHEJ repair protein recruitment at CRISPR-Cas9-induced DSBs in mouse chromocenters, suggesting a role for SMC5/6 in mammalian heterochromatin repair [21].

How SUMOylation exactly regulates DSB movement outside of the c-Het domain remains unknown. One hint for its specific role in heterochromatin repair comes from the fact that the SUMOylation activity of Drosophila Quijote and Cervantes is important to target and retain heterochromatic DSBs at the nuclear periphery [20,87]. This peripheral retention of heterochromatic DSBs depends on the presence of SUMO Targeted Ubiquitin Ligases (STUbL), which are enriched at the nuclear periphery. It is therefore hypothesized that STUbL proteins directly target or bind SUMOylated substrates at the DSB site and thereby help retain the heterochromatic DSBs at the nuclear periphery and promote safe HR repair [20] (Figure 3). Future work aimed at identifying which substrates are exactly being SUMOylated at c-Het DSBs is expected to lead to more insights into the role of SUMOylation in Rad51 exclusion as well as DSB movement in c-Het.

**Figure 3 genes-12-01415-f003:**
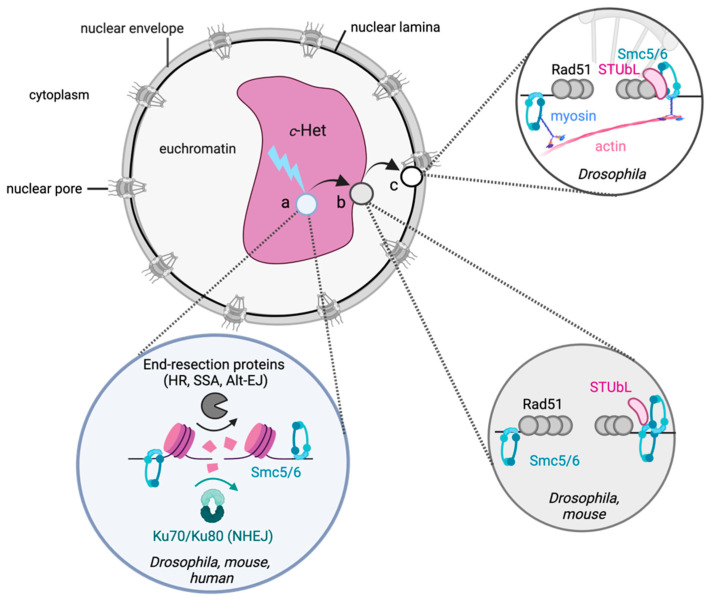
General model for DSB repair in c-Het. (**a**) DSBs in c-Het can be repaired by HR, NHEJ, single-strand annealing (SSA), and alternative end-joining (Alt-EJ) [19,21,22]. NHEJ, SSA and early HR proteins can bind to heterochromatic DSBs within the c-Het domain. (**b**) Upon commitment to HR repair, DSBs move to the periphery of the c-Het domain through the concerted actions of chromatin proteins (e.g., SMC5/6) [19], SUMOylation, and SUMO Targeted Ubiquitin Ligases (STUbL) [20,87]. At the c-Het periphery, Rad51 binds to the resected DSBs and promotes HR with homologous sequences originating from sister chromatids or homologs [22]. (**c**) Studies in Drosophila have revealed that approximately twenty percent [88] of c-Het DSBs move all the way to the nuclear periphery through myosin- and nuclear actin-mediated directed movement, where nuclear pore and inner nuclear membrane proteins directly promote HR repair of c-Het DSBs.

Nuclear F-actin and myosin were recently implicated in directly promoting the movement of c-Het DSBs to the nuclear periphery in Drosophila [88]. F-actin nucleators localize to DSBs within the heterochromatin domain, whereas F-actin fibers only form outside the heterochromatin domain, indicating that nuclear actin helps promote the directed movement of DSBs from the heterochromatin to the nuclear periphery [88] (Figure 3). Indeed, nuclear actin is emerging as an important player in nuclear architecture and repair [89] and was also recently found to help cluster DSBs in euchromatin to promote HR repair [90]. Traveling along the nuclear actin fiber presents an exciting means for fast, directed transport of the DSB away from the heterochromatin domain to promote safe repair. The SMC5/6 complex was found to promote binding of the myosin activator Unc45 to DSBs, posing the intriguing hypothesis that the SMC5/6-SUMO pathway is essential to initiate DSB movement on the actin fiber by directly promoting myosin activity [88].

What makes the nuclear periphery an effective environment for heterochromatic DSB repair? A plausible hypothesis is that the nuclear periphery could serve as an anchoring point for broken repeats to prevent aberrant recombination with identical repeats present within the heterochromatin environment. Indeed, the loss of nuclear pore or inner nuclear membrane protein complexes result in an increase in aberrant chromosomal structures, specifically involving heterochromatic regions [20]. An additional possibility is that the nuclear periphery directly guides repair pathway choice of c-Het regions by compartmentalizing specific repair proteins [91,92,93].

The dependency on the nuclear periphery for repair of heterochromatic DSBs appears analogous to the repair of persistent DSBs or collapsed replication forks in budding yeast, which also require the SUMOylation pathway to coordinate movement to the nuclear periphery [94]. Although the movement of heterochromatic DSBs to the nuclear periphery has not been described in mammalian cells to date, two recent studies have revealed that stressed-replication foci and dysfunctional telomeres can associate with the nuclear periphery in human cells [95,96]. Moreover, nuclear pore stability was found to be important to prevent telomere fragility, indicating the essentiality of the nuclear periphery in the maintenance of repetitive telomeric regions in humans [95]. It is also conceivable that mammalian cells have adopted other nuclear regions for the repair of c-Het DSBs. For example, telomeres in cancer cells that depend on an HR-type mechanism for their maintenance, termed alternative lengthening of telomeres (ALT), have been found to co-localize with promyelocytic leukemia (PML) bodies, which are membraneless, phase-separated nuclear bodies implicated in many nuclear processes [97,98,99]. Strikingly, this ALT telomere–PML body interaction has also been shown to depend on the SMC5/6–SUMO pathway [99,100].

Altogether, a picture emerges in which the nuclear periphery plays an evolutionary conserved role in safeguarding ‘difficult to repair’ regions, such as repetitive DNA or persistent DSBs, and that SUMOylation plays an important role in this process (discussed in [92,101]).

In conclusion, studies in model organisms, in particular Drosophila and mouse, have revealed the precise coordination of repair in the c-Het domain; from local chromatin changes to the spatiotemporal regulation of HR repair and directed DSB movement to the nuclear periphery. Future challenges entail understanding how all these activities are intertwined in space and time, which will lead to a deeper understanding of the intricate mechanisms required for the maintenance of the repetitive landscape of constitutive heterochromatin.

## 3. Facultative Heterochromatin

Another type of densely packaged heterochromatin is facultative heterochromatin (f-Het) (Figure 1). The level of compaction and corresponding gene silencing of f-Het depends on the developmental stage and cell type and is therefore called *facultative*, from the Latin word “facultas”, which means “opportunity”. f-Het can cover an entire chromosome (e.g., the inactive X chromosome in female mammals) [102], large genomic distances (e.g., developmental genes such as the *HOX* gene) [103], or regulatory regions (e.g., promoters). This type of chromatin is enriched for the histone mark H3K27me3 as well as polycomb proteins. Polycomb-bound chromatin tends to accumulate into nuclear foci called ‘polycomb bodies’ in flies [104] as well as human cells [105] and mainly contains silenced genes [106,107].

The first polycomb protein, Polycomb (Pc), was discovered more than 70 years ago in Drosophila [108]. The loss of Pc led to ectopic expression of the *HOX* genes, resulting in the transformation of body segments [103,108]. Most of the other polycomb-group (PcG) proteins were discovered in the 1980s and were defined as genes whose mutations yielded a similar or enhanced Pc mutant phenotype [109,110]. In Drosophila, PcG proteins form two types of complexes: polycomb repressive complex 1 and 2 (PRC1 and PRC2), which, together, are essential for silencing of developmental genes. The PRC2 complex promotes the trimethylation of histone H3 lysine 27 through the enzymatic activity of its complex member E(z) [111,112]. H3K27me3 is essential for the repression of gene transcription [113,114] and is bound by polycomb (Pc), a PRC1 complex member. The PRC1 complex on its turn promotes the ubiquitylation of H2AK118 (K119 in mammals) [115], which provides a binding site for the PRC2 complex, thereby creating a positive feedback loop to establish H3K27me3 domains [116] (Figure 1). PcG proteins can directly silence transcription by, for example, preventing the binding of chromatin remodeling complexes [117]. In addition, polycomb proteins can indirectly block transcription through a combination of local chromatin compaction [118,119,120,121], loop formation [106,107], as well as phase separation [24,28,122].

Although f-Het covers a substantial part of the genome, it remains relatively understudied in its response to DSBs. Since f-Het is important in many biological processes, such as X-chromosome inactivation in female mammals, genomic imprinting, stem cell maintenance, and cell differentiation, it is not surprising that polycomb misregulation can lead to diseases such as cancer [123]. It is, therefore, essential to understand how this silenced state is maintained during the repair of DSBs.

### DSB Repair in f-Het

Thus far, a few studies in mammalian cultured cells have investigated the repair of DSBs in H3K27me3-enriched regions. The irradiation of the inactive X chromosome in a human female fibroblast cell line resulted in the decondensation of the f-Het domain [124]. Additionally, visualizing DSB repair proteins (such as 53BP1 and RPA) at specific timepoints following ion irradiation revealed the ‘bending’ of these proteins around the inactive X chromosome, which suggests that DSBs move away from the more compact part of the inactive X chromosome [124]. If, and how, DSBs relocate in polycomb chromatin remains to be investigated, but this study does suggest that f-Het regions may require a specific repair response at DSBs.

A more recent study investigated the repair of CRISPR-Cas9-induced DSBs at imprinted alleles of mouse cells [125]. Inducing DSBs at imprinted genes is an elegant system to study the effect of chromatin state on DSB repair, since the active and silent alleles have identical DNA sequences as well as chromosomal position, ruling out DNA sequence or chromosome-specific effects on repair. Any changes in DSB repair are therefore the result of changes in the chromatin landscape. The silencing of imprinted loci occurs through a combination of different chromatin modifications including DNA methylation, H3K27me3 and H3K9me2/me3 [126]. Imprinted loci therefore slightly deviate from the f-Het definition described above. Nevertheless, inducing DSBs at imprinted loci in mouse embryonic stem cells, although resulting in a delay in the accumulation of mutations at the silenced allele due to decreased Cas9 accessibility, did not affect repair pathway usage when compared to the active allele. The ratio of c-NHEJ versus HR repair, the size of the insertions and deletions at the repaired site, as well as the frequency of repair products remained unaffected by imprinting [125]. This suggests that DSBs in silenced, imprinted loci do not require differential DSB repair pathway usage.

In contrast, a recent study that used a sequencing-based reporter screen to investigate the impact of chromatin context on CRISPR-Cas9-induced DSBs did reveal differences in repair pathway usage in H3K27me3 regions [127]. This reporter was randomly integrated in >1000 different locations in the genome of human cancer cells and can be cleaved by Cas9. The sequencing of repair products revealed a relative decrease in c-NHEJ repair products and an increase in microhomology mediated end-joining (MMEJ) in H3K27me3-enriched regions. MMEJ is a type of alternative EJ that depends on DSB end-resection and microhomologies to repair DSBs [5]. Interestingly, inhibition of the H3K27me3 methyltransferase EZH2 increased c-NHEJ repair pathway usage in H3K27me3-enriched regions to levels similar as in euchromatin, suggesting that H3K27me3 chromatin might normally be refractory to c-NHEJ repair. What exactly causes the decrease in c-NHEJ in H3K27me3-enriched regions remains unknown. However, recent studies do implicate polycomb proteins in the regulation of replication fork stability [128] as well as DNA damage repair [129]. This indicates that f-Het components could directly impact repair protein recruitment and thereby potentially shift repair pathway balance. Future studies using model systems with well-defined f-Het domains, such as the inactive X chromosome in female mammals or polycomb bodies in Drosophila animals, could reveal new insights into the repair pathway choice and spatiotemporal dynamics of DSBs in this distinct domain.

## 4. Lamina-Associated Domains

Heterochromatic sequences can be associated with the nucleolus, chromocenters, or nuclear lamina (Figure 1a). The nuclear lamina spans the inside of the nuclear membrane in metazoan cells and mainly consists of type V intermediate filament proteins lamin A and lamin B. The nuclear lamina is connected to the nuclear membrane through the lamin B receptor, whose N-terminal end interacts with lamin B, while its C-terminal end resides within the nuclear membrane [130]. Genome-wide mapping analyses have uncovered the recurrent localization of specific genomic regions to the nuclear lamina [131,132,133,134]. These regions are called lamina-associated domains (LADs) and are enriched for the proteins lamin A, B1, and B2 [135]. The composition of these LADs varies between cells, but generally consists of regions with low gene density, enrichment of silencing histone marks, such as H3K9me2/3 and H3K27me3, and repressed gene activity [132,135]. As discussed above, ‘difficult to repair’ genomic regions move to the nuclear periphery to continue repair in multiple organisms, but how the pre-existing peripheral localization of DNA could influence DSB repair pathway choice is just starting to be elucidated.

### DSB Repair in LADs

A recent study in human cancer cells has revealed that genomic regions associated with the nuclear lamina could potentially undergo differential DSB repair pathway choice [127]. Generally, DSB repair in silenced, heterochromatic regions was found to be associated with an increased usage of MMEJ and a reduction in c-NHEJ. However, regions that contain ‘triple-heterochromatin’ features, including high H3K9me2 levels, late replication, and lamina association, were more strongly associated with increased MMEJ usage compared to regions associated with either high H3K9me2 enrichment or late replication alone [127]. This suggests that lamina association somehow promotes the usage of MMEJ. This increase in alternative DSB-repair pathway usage is in line with an earlier report that found an increased use of alternative end-joining when targeting a DSB to the nuclear lamina [136]. However, the loss of lamin A or the lamin B receptor did not clearly affect repair pathway usage at CRISPR-Cas9-induced DSBs in ‘triple heterochromatin’ regions [127], indicating that lamins might not be the dominant factor in promoting MMEJ. Other features present at ‘triple heterochromatic sites’, such as the specific type of chromatin proteins or DNA sequences associated with the nuclear lamina, could play a more important role in repair pathway choice.

The decreased usage of c-NHEJ in H3K9me2-enriched heterochromatic regions identified in this study contrasts with a previous study that identified similar levels of c-NHEJ at single DSBs induced in either eu- or heterochromatic regions in Drosophila tissue [22]. These different outcomes might reflect inherent differences between species in terms of DSB repair in heterochromatic regions, but could also reflect distinctions in the exact type of heterochromatin in which the DSBs are induced. Indeed, further research will be necessary to elucidate which factors exactly contribute to DSB repair pathway choice in silenced heterochromatic LADs, and the impact thereon by different heterochromatic properties, such as (repetitive) DNA content and heterochromatin protein composition. Additionally, silenced DNA sequences are thought to stochastically reside in different heterochromatic nuclear compartments, alternating between the nuclear lamina and nucleolus or pericentromeric bodies [137]. It would be interesting, in the future, to assess the impact of these different compartments on DSB repair using single-cell methods and directly compare the response in different species.

## 5. Summary and Perspectives

Here, we reviewed the current state of the art in our understanding of the regulation of DSB repair in silenced heterochromatic regions. Collectively, these studies indicate the differential regulation of DSB repair in heterochromatic regions, in particular c-Het regions. Distinct DSB movements, spatiotemporal HR pathway regulation, and local chromatin changes drive the ‘safe’ repair of these repetitive DNA sequences at the c-Het or nuclear periphery. The eukaryotic genome consists of many more repetitive genomic regions, which often do not reside solely in silenced chromatin. For example, major parts of the repetitive ribosomal DNA are heavily transcribed [138] and centromeric repeats contain specific histone marks [139], which are different from their surrounding pericentromeric c-Het regions. Indeed, these repetitive regions have been described to have distinct repair properties [21,140,141,142], which are different from repair of repetitive sequences in silenced heterochromatin. A picture emerges where each chromatin domain in the eukaryotic nucleus requires specific repair mechanisms, such as differential chromatin changes and repair protein recruitment. In line with this, work in human cancer cells [143] and Drosophila [81] has revealed the appearance of different chromatin signatures at DSBs in ‘active’ versus ‘silenced’ chromatin domains. Future work will have to elucidate, on a genome-wide scale, which chromatin signatures appear at DSBs in distinct chromatin domains, and how this directs repair.

Chromatin signatures are extremely plastic and undergo changes upon cell differentiation, ageing as well as during tumorigenesis. It is likely that these chromatin changes will have a major impact on the efficiency of DNA repair. For example, oncometabolite-induced changes in chromatin signatures are directly linked to DSB-repair defects in cancer [144]. Understanding how chromatin changes in development, but especially in cancer, influence the stability of the underlying sequence is essential to understand the occurrence of mutational signatures in different tissues, cell types [145], and cancer types [146]. More importantly, this knowledge could, in the long-term, be used to specifically target certain cancer types, for example, by combining radiotherapy with inhibitors of specific chromatin modifiers.

The use of model organisms, with their well-defined chromatin domains and genomically stable cells, will be invaluable in determining the impact of chromatin changes on DSB repair, especially in the context of developmental- or ageing-induced [147] changes in chromatin domains. In the long-term, an improved understanding of how different pre-existing chromatin regions respond to DNA damage will undoubtedly lead to new insights into the maintenance of genome stability throughout normal organismal development as well as defects in these processes during disease onset.

## Figures and Tables

**Figure 1 genes-12-01415-f001:**
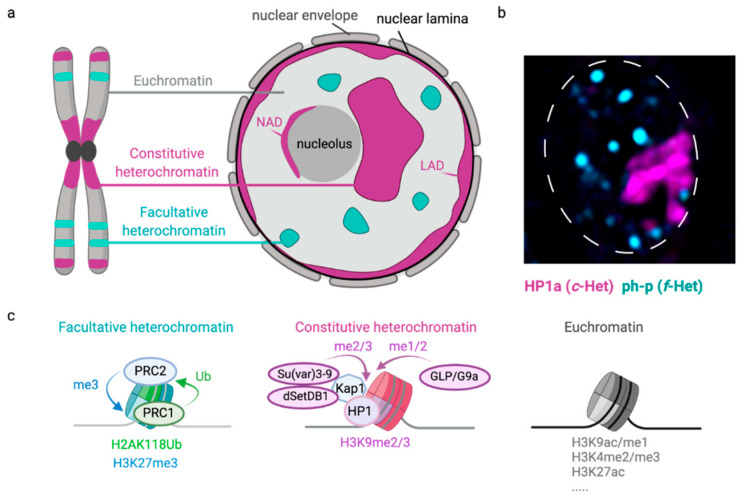
Chromatin domains in the eukaryotic nucleus. (**a**) The eukaryotic nuclear environment can be roughly divided into two types of chromatin: euchromatin and heterochromatin. Euchromatin contains many active genes and is associated with open chromatin structures. In contrast, heterochromatin is transcriptionally silent, encodes fewer genes, and has a compact, dense conformation. The two most prominent types of heterochromatin are *facultative* heterochromatin (f-Het) and *constitutive* heterochromatin (c-Het). f-Het and euchromatin are interspersed over the linear chromosome, whereas c-Het is mainly enriched around centromeres and telomeres. f-Het is associated with the transcriptional silencing of developmental genes, and often forms multiple dense polycomb bodies without membranes in the nucleus. c-Het contains many repetitive sequences and, depending on the cell type or organism studied, localizes to the nuclear lamina (lamina-associated domains, LADs), nucleolus (nucleolus-associated domains, NADs), and/or coalesces into one or a few big domains (chromocenters). (**b**) Image of a nucleus in a third instar Drosophila larval wing disc expressing fluorescently tagged ph-p (f-Het protein, PRC1 complex member, cyan) and fluorescently tagged HP1a (c-Het protein, magenta). (**c**) The basic structural unit of chromatin is the nucleosome, which consists of ~147 base pairs of DNA wrapped around eight histone proteins. Each chromatin environment in the nucleus is characterized by the presence of specific post-translational histone modifications and the recruitment of certain chromatin proteins. f-Het is enriched for H3K27me3 and H2AK118Ub in Drosophila (K119 in human). The PRC2 complex mediates the tri-methylation of Histone H3 lysine 27. The PRC1 complex binds to H3K27me3 and can ubiquitylate H2AK118, which, in turn, provides a binding motif for the PRC2 complex. c-Het is enriched for H3K9me2/me3, which, in Drosophila, is established by the methyltransferases G9a, Su(var)3-9 and dSETDB1 (eggless). H3K9me2/me3 recruits heterochromatin protein 1 (HP1), which can oligomerize and thereby create a compact, phase-separated domain. Kap1 is an HP1 binding protein and a canonical c-Het component in mammalian cells. Euchromatin is mainly enriched for histone modifications associated with active transcription, such as di- and trimethylation of H3K4 and acetylation of H3K9 and H3K27, together creating a chromatin environment that has a more open conformation and is permissive to transcription.

**Figure 2 genes-12-01415-f002:**
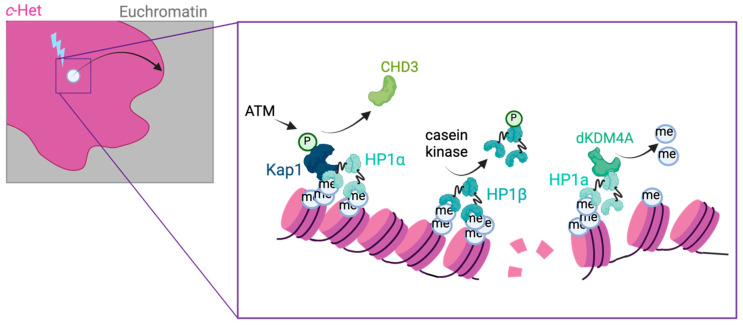
Local chromatin changes at c-Het break sites in Drosophila and mouse. DSBs in c-Het require specific chromatin modifying activities for their repair. Studies in both Drosophila and mouse cells have provided insight into the local chromatin changes required for c-Het DSB movement and repair. In mouse cells, several DNA damage-specific phosphorylation events were found to promote DSB repair in c-Het. The DNA damage kinase ATM can directly phosphorylate the heterochromatin protein Kap1 [70,73]. This phosphorylation promotes the release of the nucleosome remodeler CHD3, thereby allowing c-Het relaxation [71]. In addition, upon DNA damage, casein kinase 2 (CK2) phosphorylates HP1β, thereby promoting its release from heterochromatin and subsequent chromatin expansion at break sites [72]. In Drosophila, the enzymatic activity of the Drosophila histone demethylase dKDM4A is specifically required for c-Het repair by promoting the demethylation of the canonical c-Het histone mark H3K9me2/me3 as well as H3K56me3 [80,81]. This demethylation allows DSB movement and timely repair of c-Het DSBs.

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
