# Peer review of "The Sound of Silence: How Silenced Chromatin Orchestrates the Repair of Double-Strand Breaks"

_genes, 2021, doi:10.3390/genes12091415_

Round 1

Reviewer 1 Report

This manuscript nicely reviews the processes specific to the DSB repair of silenced regions of the genome. The text introduces each type of the silenced domains (constitutive and facultative heterochromatin and lamina-associated domains) and then presents the peculiarities of DSB repair there.  
The authors focus on the roles of heterochromatin proteins and the principles of DSB repair in heterochromatic domains. The latter mainly concerns the movement of the lesion to nuclear domains (outside of heterochromatin) that would disfavor illegitimate recombination events, and the mechanisms underlying it. The authors also present the local chromatin changes during DSB repair in constitutive heterochromatin and the evidence that H3K27me3-enriched regions may need a specific DSB repair response. Finally, the authors address the repair pathway usage in lamina-associated domains.
Overall, this review is an excellent way to introduce someone already familiar with the mechanisms of DSB repair with the specifics of the process in silenced regions of the genome. 
I have no major issues with this manuscript. It is well structured, clearly written, and informative. Still, the authors might consider the following minor issues in improving the text:
1. To me, the overemphasis on justifying the use of model organisms (lines 14 - 16, lines 76-78, lines 497 - 498) seems somewhat unnecessary. I would have omitted the sentence: "Because of their well-characterized chromatin domains and genomically stable cells, model organisms provide a valuable tool to dissect the impact of chromatin on DNA damage repair." (lines 14-16) from the abstract. 
2. The sentence: "When repaired improperly, DSBs can result in minor changes to DNA sequences and the formation of major chromosomal rearrangements, such as translocations, dicentric chromosomes or chromothripsis." (lines 26-28) could probably be rewritten as putting minor sequence changes, and changes to chromosome structure together could confuse the reader.
3. While the authors list the functional homologues of H2AX in various organisms (line 51), they limit further discussion of histone modifications and chromatin remodelling in DSB repair to "many other chromatin changes occur at the break site that contribute to the faithful execution of repair." I would suggest either to expand slightly on the topic or altogether leave out the sentence: "One of the first chromatin events at DSBs is the phosphorylation of the histone variant H2A.X (gH2A.X in mammals, gH2A in yeast, gH2A.v in Drosophila), which initiates a multitude of downstream repair events."
4. On line 46, "to be able to" is unnecessary.
5. On line 66, is the word "acquiring" right (or needed) in the context.

Reviewer 2 Report

In the manuscript „The sound of silence: how silenced chromatin orchestrates the repair of double-strand breaks“ A Kendek, M Wensveen and A Janssen summarize our current understanding of DNA repair in both H3K9me2/me3 and H3K27me3 marked heterochromatin. Overall the review is well written and provides a good summary of a complicated, but highly relevant topic. I have no major concerns that would prevent a publication. However, there are several minor points that I think will facilitate the understanding of the article:

I would further encourage the authors to modify the figures as follows:

Figure 1c: please add dSetBD1, KAP1 and G9a to the scheme and capitalize Su(var)3-9. This would ensure that all chromatin components mentioned in the manuscript are represented in the Figure

Figure 2: It would be helpful to expand this figure into an illustration to encompass the different repair pathways (HR, NHEJ, MMEJ with the proteins mentioned in the review and under which circumstances/organism the pathway has been observed).

Comments to the text:

(In order of appearance)

  1. Page 1 Line 14-17: formatted in a different font and size
  2. Page 1 Line 26-27: Minor changes implies a judgmental statement. Please change to: "When repaired improperly, DSB can result in defects ranging from small indels to the formation of major chromosomal rearrangements, such as..."
  3. Page 2 Line 63: Even though the review is focused on heterochromatin I think it would be helpful to mention that RNA Pol 2 removal seems to be important for DSB repair, as it represents the major feature of euchromatin that is missing in heterochromatin. e.g.:

Caron P, Pankotai T, Wiegant WW, et al. WWP2 ubiquitylates RNA polymerase II for DNA-PK-dependent transcription arrest and repair at DNA breaks. Genes Dev. 2019;33(11-12):684-704. doi:10.1101/gad.321943.118

Shanbhag NM, Rafalska-Metcalf IU, Balane-Bolivar C, Janicki SM, Greenberg RA. ATM-dependent chromatin changes silence transcription in cis to DNA double-strand breaks. Cell. 2010;141(6):970-981. doi:10.1016/j.cell.2010.04.038

Poli J, Gerhold CB, Tosi A, et al. Mec1, INO80, and the PAF1 complex cooperate to limit transcription replication conflicts through RNAPII removal during replication stress. Genes Dev. 2016;30(3):337-354. doi:10.1101/gad.273813.115

  1. Page 2 Line 64: consider rewording for a better understanding: "...striking movements of the DSB to the periphery of the heterochromatic compartment, or the nuclear periphery to repair the damage...".
  2. Page 2 Line 72: please provide a reference to a review giving an overview of repetitive elements.
  3. Page 2 Line 89-90: given the redundancy between Su(var)3-9, G9a, and dSETDB1 please be more specific: Su(var)3-9 is the major HMT for H3K9me3 at pericentric heterochromatin, were it acts partially redundant with dSETDB1. This would also be the place to introduce KAP1 as it recruits SETDB1 and is mentioned but not introduced later. e.g.:

Brent Brower-Toland, Nicole C Riddle, Hongmei Jiang, Kathryn L Huisinga, Sarah C R Elgin, Multiple SET Methyltransferases Are Required to Maintain Normal Heterochromatin Domains in the Genome of Drosophila melanogaster, Genetics, Volume 181, Issue 4, 1 April 2009, Pages 1303–1319, https://doi.org/10.1534/genetics.108.100271

  1. Page 2 Line 92: please include HP1b here. As it localizes to heterochromatin as well. Please also add a short sentence about HP1alpha/beta as these are mentioned but not introduced later.

e.g.:

Smothers JF, Henikoff S. The hinge and chromo shadow domain impart distinct targeting of HP1-like proteins. Mol Cell Biol. 2001;21(7):2555-2569. doi:10.1128/MCB.21.7.2555-2569.2001

  1. Page 2 Line 100: It would be helpful to mention here that across species the underlying sequence of pericentric heterochromatin is dominated by satellite repeats and scattered transposons. Please also include a brief statement that highlights what these sequences are and why repairing them might be relevant.
  2. Page 3 Line 107: Please be more specific. Heterochromatin organization in Drosophila can as well be in several domains e.g. (Strom et al., 2017), the organization of pericentric heterochromatin in salivary gland cells is also termed "chromocenter" and in mice the organization of pericentric heterochromatin and the number of foci is highly dependent on the cell type (e.g. Solovei et al., 2013).

Strom AR, Emelyanov AV, Mir M, Fyodorov DV, Darzacq X, Karpen GH. Phase separation drives heterochromatin domain formation. Nature. 2017;547(7662):241-245. doi:10.1038/nature22989

Solovei I, Wang AS, Thanisch K, et al. LBR and lamin A/C sequentially tether peripheral heterochromatin and inversely regulate differentiation. Cell. 2013;152(3):584-598. doi:10.1016/j.cell.2013.01.009

  1. Page 4 Line 174-175: please introduce ATRIP and Rad51 with a very short description
  2. Page 5 Line 182: Please replace Towbin et al, with Padeken et al., 2019 as Towbin et al does not analyse genome stability, while Padeken et al link satellite repeat expression to BRCA1 and DNA damage induced apoptosis in the same model organism.

Padeken J, Zeller P, Towbin B, et al. Synergistic lethality between BRCA1 and H3K9me2 loss reflects satellite derepression. Genes Dev. 2019;33(7-8):436-451. doi:10.1101/gad.322495.118

  1. Page 5 Line 193: please shortly introduce SMC5/6.
  2. Page 10 Line 449: It would be helpful to add a sentence about the proteins known to mediate the interaction between heterochromatin and the nuclear periphery to put the lamin A / LBR data into perspective.
  3. Page 11 Line 465: Please add Kind et al., 2013 as a reference, as they have directly visualized the LAD dynamics over time.

Kind J, Pagie L, Ortabozkoyun H, et al. Single-cell dynamics of genome-nuclear lamina interactions. Cell. 2013;153(1):178-192. doi:10.1016/j.cell.2013.02.028
